# Influence of Interpenetrating Chains on Rigid Domain Dimensions in Siloxane-Based Block-Copolymers

**DOI:** 10.3390/polym14194048

**Published:** 2022-09-27

**Authors:** Stepan A. Ostanin, Maxim V. Mokeev, Vjacheslav V. Zuev

**Affiliations:** 1ITMO University, Kronverkskiy pr. 49, 197101 Saint-Petersburg, Russia; 2Institute of Macromolecular Compounds of the Russian Academy of Sciences, Bolshoi pr. 31, 199004 Saint-Petersburg, Russia

**Keywords:** coating, nanostructure, nuclear magnetic resonance (NMR), morphology

## Abstract

^1^H spin-diffusion solid-state NMR was utilized to elucidate the domain size in multiblock-copolymers (BCPs) poly-(*block* poly(dimethylsiloxane)-*block* ladder-like poly(phenylsiloxane)) and poly-(*block* poly((3,3′,3″-trifluoropropyl-methyl)siloxane)-*block* ladder-like poly(phenylsiloxane). It was found that these BCPs form worm-like morphology with rigid cylinders dispersed in amorphous matrix. By using the combination of solid-state NMR techniques such as ^13^C CP/MAS, ^13^C direct-polarization MAS and 2D ^1^H EXSY, it was shown that the main factor which governs the diameter value of these rigid domains is the presence of interpenetrating segments of soft blocks. The presence of such interpenetrating chains leads to an increase of rigid domain diameter.

## 1. Introduction

The development of microelectronics has a goal to produce nanosized chips with dimensions of about 5 nm. The production of such devices is still a challenge and needs to introduce an innovative approach. The self-organization of block copolymers (BCPs) joining the benefits of traditional lithography and thermodynamics stimulated organization for spontaneously forming nanostructures with high level of perfection, accuracy, registration and complexity is an appealing choice. For microelectronic applications, the BCP patterns must be endured into other inorganic materials, which involves the using blocks must have enough etch contrast to selectively remove one block and leave a structure made from the other block at lithography [1]. The silicon-based BCPs are specifically feasible to be used as an etch mask for pattern transfer [2]. Another advantage of silica-containing blocks is the large segregation strength that leads to small feature size of forming subunits and high etch contrast for fabrication of robust well-defined nanopatterns with high resolution [3].

The ability of self-organization of BCPs is well studied both in theory [4] and experimentally [5]. Minimization of the interfacial tension in BCPs drives the system to form multiple phase behavior [6]. An alternative way to enrich phase behavior is the increase of block number in BCPs. Hence, an increase of block number in BCPs from two to three allowed to identified in linear ABC triblock BCPs over 30 different phases [7]. Therefore, the phase behavior of multiblock copolymers with well-defined blocks can be very interesting [8]. For the lithographic applications, most interesting are morphologies such as spheres or cylinders. For advanced nanopatterning applications, the possibility of obtaining such domains with sub-5 nm size would be very interesting. The factors which influence domain dimensions are Huggins interaction parameters of blocks, their molecular weight and the presence of interpenetrating chains of one block into the domains formed by other blocks [9].

The aim of the present paper is to study the factors that govern the domain formation in linear/ladder-like polysiloxane BCPs with methyl-, trifluoropropyl- and phenyl-siloxane units using combination of different solid-state NMR methods. The chosen BCPs are composed of blocks with different Huggins interaction parameters and rigidity with well-defined structure and MM of blocks [10].

## 2. Experimental

### Materials

The BCPs I–II (Figure 1) under study were synthesized previously [10]. The blocks are poly(dimethylsiloxane) (PDMS, m = 111, M_n_ = 8400), poly(methyltrifluoropropylsiloxane) (PMFS, m = 101, M_n_ = 16,000) and ladder-like poly(phenylsiloxane) (LAD, m = 15, M_n_ = 3950). The molar mass of BCPs is for I (M_w_ = 9.0 × 10^4^, M_w_/M_n_ = 1.97) and for II (M_w_ = 4.2 × 10^4^, M_w_/M_n_ = 2.32).

## 3. Methods

### 3.1. Microscopy

A Supra 55 VP scanning electron microscope (SEM) (Carl Zeiss, Germany) was used to analyze the morphology of BCPs. 

### 3.2. NMR Spectroscopy

The descriptions of NMR experiments [11,12] are given in the Appendix A.

## 4. Results and Discussion

As received, BCP I is a white-yellowish powder and II is a white rubber. Both these BCPs are well soluble in many solvents [10]. However, as is typical for BCPs, some solvents are good for one type of blocks, and others are good for another; moreover, one solvent may be good for one block and at the same time poor for another [13]. This fact opens the possibility to manipulate the morphology of casted films by using different solvents and solution state (good solution, emulsion etc.). We prepared the films of BCPs under study from different solvents (toluene, butyl acetate, chloroform, carbon tetrachloride, etc.), which differ in quality with respect to these BCPs and constituting blocks. All films were dried at 60 °C before studying.

We recorded the SEM images of casted films (Figure 1). 

As one can see, the morphology of casted films is practically the same and does not depend on used solvents. All films have a worm-like morphology with cylindrical domains, with the length varied in the range 20–100 nm. One reason for these observations can be low glass transition temperatures of PDMS or PFMS blocks, which are much lower than the room temperature at which the casted films dried. Therefore, the high segmental mobility at room temperature leads to the formation of the same pseudo-equilibrium morphology of films regardless of the casting solvent. To elucidate other factors determining film morphology, we used solid-state NMR experiments. 

First, we determined diameters of domains formed by BCPs I and II. For this purpose, we used the proton spin-diffusion NMR technique. This technique of exploiting differences of mobility in hard (below glass transition) and soft (above glass transition) regions, is known to be sensitive to the size of the supramolecular aggregates, and was successfully used by us previously [14] according the method developed in [15]. In the case under study, there are obvious differences in molecular mobility between soft PDMS or PFMS blocks and ladder-like hard block. The domain sizes of the discrete phase B (in our case, their diameter) in the two-phase A/B mixture *d_dis_* can be determined by the method suggested by Spiess et al. [16] and were described in our previous paper [14] (See Appendix A). The spin-diffusion curves are shown in Figure 2. The dependence showed a linear increase with mixing time at small values, as expected for a two-phase system with a small interfacial thickness with respect to the size of different domains. However, any deviation from linearity show the absence of full separation between the polymer chains of constructed BCPs blocks. A similar picture was observed for all studied samples of BCPs I and II casted from different solvents. The difference in mixing times was less than 10%. Hence, little difference is expected between diameters of domains formed by BCPs I and II in the films casted from different solvents. 

The values of domain diameters and long periods are presented in Table 1 for BCPs I and II samples.

As one can see, the differences in hard domain diameters are minimal. We used a number of solid-state NMR spectroscopic methods to study the chemical compositions of different phases of BCPs I and II in order to reveal the effect of BCP structure on their morphology. ^13^C CP/MAS technique gives information about the contain of regions with restricted mobility, e.g., in our case about the chemical composition of the rigid domains. Microphase-mixed and microphase-separated structures were complex separate based only on the ^13^C CP/MAS experiment. Such experiments with a short contact time (50 µs) and high power proton decoupling are more suitable for the rigid volume of multiphase materials, since cross polarization has a low sensitivity for the mobile phase due to high mobility of components of these phase. We use these conditions in our experiments. 

Direct polarization MAS experiment with low power proton decoupling provides signals only from the soft (mobile) phase. Echo detection in such experiments gives additional filtering of the rigid component. The obtained spectra are presented in Figure 3. The ^13^C CP/MAS spectrum of BCP I shows the minimal presence of PDMS chains in rigid phase. Simultaneously, some phenyl units are present in the mobile phase (Figure 3). Therefore, we observe the mixing of different blocks of I both in rigid domains and in amorphous phase. The used experiments did not allow to measure quantitatively the extent of this mixing.

The observed picture for BCP II is somewhat different (Figure 3b). The ^13^C CP/MAS spectrum of BCP II shows the presence of PFMS chains in rigid domains. However, the ^13^C direct-polarization MAS spectrum did not show any trace of aromatic units in the mobile phase. Hence, fluorine-containing PFMS chains have a more tendency to prevent the dissolution of silicon-aromatic ladder-like blocks than PDMS chains. We observed a similar effect in the case of aromatic–aliphatic polyurethanes containing perfluorinated ethylene oxide units [17]. 

To elucidate the situation with the mixing of flexible and rigid blocks in the BCPs under study, we used the solid-state two-dimensional EXSY NMR experiments. This powerful technique allows to determine the spin-exchange between atoms located no farther than 5 Å [18]. We recorded the 2D EXSY NMR spectra of the films of BCPs I and II casted from different solvents (Figure 4).

As one can see from Figure 4, 2D ^1^H MAS solid-state EXSY NMR spectra of both copolymers contain the cross-peaks between aliphatic and aromatic signals. This result shows that distances between any protons in these blocks do not exceed 5 Å. Hence, this experiment confirms the presence of interpenetrating chains of soft and ladder-like blocks or in amorphous or/and in rigid domains. However, these results did not allow the quantitative evaluation of the fraction of such chains. The relative intensity of cross-peak in the ^1^H MAS solid-state EXSY NMR spectrum of DCP II is higher than the corresponding intensity in the similar spectrum of I, but this intensity can not be directly used for both quantitative and qualitative interpretations [19]. 

For this purpose, we used the measurements of the transverse magnetization relaxation time T_2_, which gives information about mobility in rigid and soft domains and allows to calculate the relative fraction of chains with each type of mobility. We assume that these BSPs contain rigid quasi-crystalline domains and the amorphous matrix with two components: mobile and rigid. The relaxation decays of BCPs I and II well approximated by linear combination of fast-decaying Gaussian functions (for rigid domains) and two slow-decaying single-exponential functions (for amorphous matrix) are as follows:(1)$$A(t)=A_{1}\;\exp\;\exp[-{\left(\frac{t}{T_{2}^{1}}\right)}^{2}]+A_{2}\;\exp\;\exp[-{\left(\frac{t}{T_{2}^{3}}\right)}^{2}]+A_{3}\;\exp\;\exp[-\left(\frac{t}{T_{2}^{3}}\right)]$$

The restraint on the values of T_2_ (~10 ms) for mobile phase was elucidated from spin-diffusion NMR experiment (see above). Experimental data and fits from this procedure are shown in Figure 5.

Good quality fitting has been obtained (adjustable R-square of fitting is better than 0.999 for all curves as the curves). The fit lines mostly pass through the dots (experimental data). Hence, we can qualitatively interpret these results. A more complete list of the fitting results, including the fast- and slow-decaying fractions is shown in Table 2, where the fractions were normalized to 100%.

As one can see from Table 2, the fractions of rigid and amorphous phases of BCP I are in a good agreement with volume fractions of flexible and ladder-like block calculated on basis of their MM and densities (about 25 and 75%). Hence, almost complete separation of flexible and ladder-like blocks on microphase level is observed. This was also supported by ^13^C CP/MAS spectrum (see Figure 3a). However, some segments of PDMS blocks are included in rigid domains, and similarly the segments of ladder-like blocks are dissolved in amorphous matrix. 

The situation is different for BCP II. The calculated volume fraction of ladder-like blocks is about 17%. However, based on relaxation measurements, the estimated fraction is 22% (Table 2). The ^13^C CP/MAS spectrum of II (see Figure 3b) shows the presence of soft segments in rigid domains. Hence, in the case of BCP II the presence of large fraction of PFMS segments in rigid domains leads to their higher diameter in comparison with those of rigid domains I and to an increase of the fraction of such domains. A similar situation has been observed by us in a study of polyurethanes, which are multiblock copolymers with alternating hard and soft segments [17]. These BSPs form cylindrical rigid domains. As we have shown, their diameters increase if soft segments can be dissolved in rigid domains. The opposite situation is observed if such dissolution is impossible, for example in the case of perfluoroethylene oxide soft blocks. These units cannot be dissolved in the hard aromatic domains, and, as a result, the diameters of such domains decrease significantly [17]. Hence, mutual solubility of hard and soft blocks in BCPs is one the main factor which governs the dimension parameters of morphology of BCPs.

The presence of two phases with different mobilities in the soft matrix can be explained by the presence of two types of soft segments in the BCPs under study. The formation of crystallites in semi-crystalline polymers leads to formation of folded chains in amorphous zones [20]. A similar situation is observed at formation of rigid domains in BCPs [21]. These folded segments formed an amorphous phase with restricted mobility. The terminal segments form amicrophase regions with elongated chains with developed molecular mobility.

## 5. Conclusions

By using different solid-state NMR techniques, we have investigated the structural factors which govern the dimension of domains formed BCPs. In the example of multiblock-copolymers based on rigid ladder-like poly(phenylsiloxane) blocks and flexible poly(dimethylsiloxane) or 3,3′,3″-trifluoropropyl-(methyl)siloxane blocks, which form morphology with cylindrical rigid domains distributed in amorphous matrix, it was shown that the main factor that governs the diameter value of these rigid domains is the presence of interpenetrating segments of soft blocks. The presence of such interpenetrating chains leads to an increase of diameter of rigid domains. The casted films of these BSPs formed regular structures with uniform domains. These findings are important for theoretical modeling of phase behavior of BCPs and the preparation of BCPs for lithographic application with sub-5 nm dimension of units. 

## Data Availability

The datasets generated and/or analyzed during current study are available from the corresponding author upon reasonable request.

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
