# Peer review of "Influence of Interpenetrating Chains on Rigid Domain Dimensions in Siloxane-Based Block-Copolymers"

_polymers, 2022, doi:10.3390/polym14194048_

Round 1

Reviewer 1 Report

This manuscript deals with the spin-diffusion solid-state NMR investigation of various polysiloxane block copolymers in order to determine the domain sizes in such macromolecular materials. It is proposed to accept this paper after some revision on the basis of comments below.

COMMENTS  

1.

The authors discuss the NMR techniques used for the characterization of block copolymers (pages 2-4). In this context, it should be added in this manuscript that spin-diffusion solid-state NMR investigations for determination of the domain sizes were also used for conetworks composed of chemically crosslinked polymer segments. The authors are recommended to study and cite the following publication:

Domjan, A.; Erdodi, G.; Wilhelm, M.; Neidhofer, M.; Landfester, K.; Ivan, B.; Spiess, H. W. Structural studies of nanophase-separated poly (2-hydroxyethyl methacrylate)-l-polyisobutylene amphiphilic conetworks by solid-state NMR and small-angle X-ray scattering. Macromolecules 2003, 36, 9107-9114.

2.

The name of the polymers should be written correctly according to the IUPAC Nomenclature rules in the whole text. This means that correctly poly(dimethylsiloxane), poly(methyltrifluoropropylsiloxane), poly(phenylsiloxane) (with parentheses and e.g. not polydimethylsiloxane).  

3.

SEM is not a suitable measurement technique for the determination of the domain sizes in the investigated range. Scattering techniques, such as small angle X-ray scattering (SAXS) or small angle neutron scattering (SANS) measurements would be the better choice.

4.

In Table 1, what is the source of the density values. Especially the density value of 0.03 g/cm3 is extremely low, and should be erroneous. In addition, which phases belong to A and B. This should be clarified clearly in the text and in the Table caption as well. It should noted that if wrong density values are used for the evaluation of the domain sizes and long periods, then these values are incorrect.

5.

The values for d and L with three decimal values cannot be accepted in Table 1. The applied NMR method does not provide such high precisity. One decimal value (rounded) should be used.

6.

The casting process, that is, the concentration of the solutions and the drying process is not provided at all. Therefore, this manuscript doesy not meet the most important criteria of scientific publications, that is, reproducibility. The experimental details should be provided in this paper.

7.

The authors should describe whether the casted films were annealed or not.

8.

It is absolutely unclear what is the difference between BSP II (caption Figure 1) and BCP II samples.

9.

In Figure 2, the spin diffusion results are shown for BCP II samples casted from ethyl acetate and toluene. However, the SEM image of BSP II (if it is equal to BCP II) is shown only for the sample casted from toluene, and SEM image for casted BCP II sample from ethyl acetate is not provided.

Author Response

Reviewer 1.

  1. Suggestion to add in Introduction section point that spin diffusion technics used also for characterization of conetwors but not only block-copolymers. This suggestion based on paper Spiess et al. Structural studies of nanophase-separated poly (2-hydroxyethyl methacrylate)-l-polyisobutylene amphiphilic conetworks by solid-state NMR and small-angle X-ray scattering. Macromolecules2003, 36, 9107-9114.

In title of this paper presents the word “conetworcs” but the objects under study are multi-block copolymers (the graft-copolymers where both ends of grafted block united to main chain of based block). This situation is similar to our case under study, only the density of grafting in Spiess paper is higher. The methodology in paper of Spiess are the same as in other his papers cite by us. Hence, new citation is superfluously because basic papers of Spiess are cited.

  1. The name of the polymers should be written correctly according to the IUPAC Nomenclature rules in the whole text. This means that correctly poly(dimethylsiloxane), poly(methyltrifluoropropylsiloxane), poly(phenylsiloxane) (with parentheses and e.g. not polydimethylsiloxane).  

We accept this and repair our paper.

  1. SEM is not a suitable measurement technique for the determination of the domain sizes in the investigated range. Scattering techniques, such as small angle X-ray scattering (SAXS) or small angle neutron scattering (SANS) measurements would be the better choice.

We agree with this comment. However, the SANS is practically not available and for such measurements need deuterium labeling of one block, that is practically complex to realization.

  1. In Table 1, what is the source of the density values. Especially the density value of 0.03 g/cm3 is extremely low, and should be erroneous. In addition, which phases belong to A and B. This should be clarified clearly in the text and in the Table caption as well. It should noted that if wrong density values are used for the evaluation of the domain sizes and long periods, then these values are incorrect.

The calculation of proton densities  was made according F. Mellinger, M. Wilhelm, and H. W. Spiess. Calibration of 1H NMR Spin Diffusion Coefficients for Mobile Polymers

through Transverse Relaxation Measurements Macromolecules 1999, 32, 4686-4691

The values of o.03 g/cm3 is normall. The mistake we made with 0.9 g/cm3, where we missed zero. Should be 0.095.      We accept the suggestion to round the d and L values to one decimal value.

  1. The authors should describe whether the casted films were annealed or not.

We added this description

  1. It is absolutely unclear what is the difference between BSP II (caption Figure 1) and BCP II samples.

The difference is absence. The aim of these images is to show that only NMR can found the difference in the morphology.

  1. In Figure 2, the spin diffusion results are shown for BCP II samples casted from ethyl acetate and toluene. However, the SEM image of BSP II (if it is equal to BCP II) is shown only for the sample casted from toluene, and SEM image for casted BCP II sample from ethyl acetate is not provided.

It was made because the results obtained for films casted from different solvent are very similar.

Reviewer 2 Report

The authors discussed the domain sized in block copolymers. This is a really interesting topic for MDPI polymer readers and the authors have presented the results effectively. However, I failed to understand how different sizes and morphology influence microelectronic applications.  These require a high degree of accuracy but the BCPs presented are random. Finally, the conclusions are vague without many details in it. 

Author Response

Reviewer 2.

As you can see from Figure 1 (a-c) the casted films formed regular structure with uniform domains. It is so what need for lithographic application. Random is structure of BCP but morphology is regular.

We increase Conclusions.
